# Characterization of Ocular Clinical Isolates of *Pseudomonas aeruginosa* from Non-Contact Lens Related Keratitis Patients from South India

**DOI:** 10.3390/microorganisms8020260

**Published:** 2020-02-15

**Authors:** Alpana Dave, Apurwa Samarth, Roshni Karolia, Savitri Sharma, Esther Karunakaran, Lynda Partridge, Sheila MacNeil, Peter N. Monk, Prashant Garg, Sanhita Roy

**Affiliations:** 1Prof. Brien Holden Eye Research Centre, LV Prasad Eye Institute, Hyderabad 500034, India; dave.alpana@gmail.com (A.D.); apurwasamarth25@gmail.com (A.S.); prashant@lvpei.org (P.G.); 2Jhaveri Microbiology Centre, LV Prasad Eye Institute, Hyderabad 500034, India; rkroshnikaroliya@gmail.com (R.K.); savitri@lvpei.org (S.S.); 3Department of Chemical and Biological Engineering, University of Sheffield, Sheffield S102TG, UK; e.karunakaran@sheffield.ac.uk; 4Department of Molecular Biology and Biotechnology, University of Sheffield, Sheffield S102TG, UK; l.partridge@sheffield.ac.uk; 5Department of Material Science and Engineering, University of Sheffield, Sheffield S102TG, UK; s.macneil@sheffield.ac.uk; 6Department of Infection Department of Infection, Immunity and Cardiovascular Disease, University of Sheffield, Sheffield S102RX, UK; p.monk@sheffield.ac.uk

**Keywords:** type III secretion, antibiotic resistance, Pseudomonas, biofilm, pyoverdine, swarming

## Abstract

*P. aeruginosa* is the most common Gram-negative organism causing bacterial keratitis. *Pseudomonas* utilizes various virulence mechanisms to adhere and colonize in the host tissue. In the present study, we examined virulence factors associated with thirty-four clinical *P. aeruginosa* isolates collected from keratitis patients seeking care at L V Prasad Eye Institute, Hyderabad. The virulence-associated genes in all the isolates were genotyped and characteristics such as antibiotic susceptibility, biofilm formation, swarming motility, pyoverdine production and cell cytotoxicity were analyzed. All the isolates showed the presence of genes related to biofilm formation, alkaline proteases and elastases; however, there was a difference in the presence of genes related to the type III secretion system (T3SS). A higher prevalence of *exoU+* genotype was noted in the drug-resistant isolates. All the isolates were capable of forming biofilms and more than 70% of the isolates showed good swarming motility. Pyoverdine production was not associated with the T3SS genotype. In the cytotoxicity assay, the presence of *exoS,*
*exoU* or both resulted in higher cytotoxicity compared to the absence of both the genes. Overall, our results suggest that the T3SS profile is a good indicator of *P. aeruginosa* virulence characteristics and the isolates lacking the effector genes may have evolved alternate mechanisms of colonization in the host.

## 1. Introduction

*P. aeruginosa*, a Gram-negative bacterium, is ubiquitous in nature and a major opportunistic human pathogen. It is one of the most common causative agents for bacterial keratitis in India and worldwide [1]. *P. aeruginosa* adheres to the cell surface and releases toxins that result in recruitment of inflammatory cells leading to corneal scarring [2,3] that may lead to perforation of the cornea within 48–96 h of infection [4]. Contact lens wearers are at a higher risk of developing keratitis in developed countries, while ocular trauma and injury are the major risk factors in developing countries [5]. *P. aeruginosa* also causes acute or chronic infections in patients with cystic fibrosis, cancer or extensive burns [6]. It has a repertoire of virulence factors such as presence of flagellin and type IV pili along with secreted exotoxins, proteases and elastases. A combination of these factors determines the ability of an isolate to invade the host cell and colonize. One important virulent mechanism is the type III secretion system (T3SS) which directly injects effector proteins into the host cells [7]. ExoS, ExoT, ExoY and ExoU are the four effector enzymes that are main focus of research. ExoS and ExoT are closely related bifunctional enzymes with Rho-GAP and ADP-ribosyltransferase activities [8]. While ExoU is a potent phospholipase, ExoY functions as adenylate cyclase [9,10]. The T3SS regulon consists of five operons including *pscL* and *pscU* that encodes components of secretion machinery [11]. Another virulence associated factor is the swarming ability which is attributed to its rotating polar flagellum. A study by Overhage et al. found that two virulence genes, *lasB* and *pvdQ* were required for swarming motility and also that swarmer cells exhibit increased antibiotic resistance [12]. Swarming also helps to prevent phagocytosis of the bacteria by host cells [13]. *P. aeruginosa* also secretes several extracellular proteases like alkaline protease (AprA), elastase (LasB) and protease IV (PrpL). While both AprA and LasB are metalloproteases, PrpL is serine protease in nature and all of these proteases has been reported to play a role in corneal infections [14,15,16]. AprA has been shown to impede bacterial clearance by host cells by preventing complement mediated phagocytosis, similarly, LasB degrades mucins and surfactant proteins that aids in bacterial clearance [17,18]. Protease IV is a key virulence factor of *P. aeruginosa* and is induced by quorum sensing. *Pseudomonas* spp. are also known to form biofilms that prevent the penetration of antibiotics contributing to its virulence and are notoriously difficult to eradicate [19]. *LadS*, a calcium-responsive kinase is required for biofilm formation and is responsible for the swirtch in acute-chronic *Pseudomonas* infection [20,21]. Pyoverdine, a siderophore produced by *Pseudomonas* spp., is also known to contribute to its virulence. This plays an important role in chelating iron from the host or the environment and also imparts a green fluorescence [22]. A combination of these virulence factors facilitates infection and may confer antibiotic resistance to the bacteria. Subedi et al. reported that levofloxacin, ciprofloxacin and amikacin were the most effective drugs for ocular infections [23]. They found that the antibiotic resistance rates in ocular isolates have been stable [24] however, a recent report from Das et al. found a significant decrease in susceptibility in *Pseudomonas* spp. isolated from keratitis patients to a fourth generation fluoroquinolone, moxifloxacin [25], suggesting a rise in antibiotic resistance in *Pseudomonas* spp.

Various studies have investigated the virulence factors in clinical isolates from diseases such as cystic fibrosis, respiratory infections, septicemia, and keratitis [26,27,28,29,30]. However, with increased antibiotic resistance in the strains, an update on the virulence factors associated with *P. aeruginosa* corneal infections is warranted. In the present study, we examined the clinical features and virulence factors associated with thirty-four *P. aeruginosa* isolates from keratitis patients with non contact lens related ocular infections.

## 2. Materials and Methods

### 2.1. Bacterial Culture

Thirty-four clinical isolates were obtained from Jhaveri Microbiology Centre, LV Prasad Eye Institute, and two laboratory strains, PAO1 and PA14 (a kind gift from Dr. Urs Jenal, University of Basel, Basel, Switzerland) were used in this study and approved by the Institutional Review Board. For clinical isolates, corneal ulcer scrapings collected aseptically were investigated for bacterial and identification, following the Institute protocol as described earlier [31]. Briefly, ulcer scrapings were placed on two glass slides (Gram stain and 10% potassium hydroxide with 0.1% calcofluor white) for direct microscopy and also inoculated in different specific media for bacterial cultures. Only significant isolates as per the defined criteria were included in the study [32]. The pure homogenous culture was then subjected to VITEK^®^ 2 compact (bioMerieux, France) analysis for identification of the bacterium alongside Gram stain and series of biochemical tests. All strains of *P. aeruginosa* were grown as described earlier [33]. In brief, bacteria were sub-cultured from overnight culture in Luria Bertani media (MP Biomedicals, Mumbai, India), washed twice in 1X PBS, centrifuged at 10,000 rpm for 5 min, and resuspended in 1X PBS. Dilutions of the sample were done with serum-free media for the final inocula.

### 2.2. Antibiotic Susceptibility Test

For antibiotic susceptibility testing, minimum inhibitory concentration (MIC) was determined using Ezy MIC^TM^ strips (Himedia Laboratories, Telangana, India) or VITEK^®^ 2 AST cards according to manufacturer’s protocol following CLSI guidelines [34]. The isolates were screened for susceptibility towards chloramphenicol, fluoroquinolones such as ciprofloxacin, moxifloxacin, gatifloxacin, ofloxacin and levofloxacin, aminoglycosides such as gentamycin, amikacin and tobramycin, polymyxins such as colistin, cephalosporins such as ceftazidime and cefepime, carbepenems such as imipenem, doripenem and meropenem, glycycline such as tigercycline and ureidopenicillins and β-lactam inhibitors such as piperacillin/tazobactum, ticarcillin/clavulanic acid and cefoperazone/subalactam. Multi-drug resistance (MDR) was defined as acquired non-susceptibility to at least one agent in three or more antimicrobial categories, extensive drug resistance (XDR) was defined as non-susceptibility to at least one agent in all but two or fewer antimicrobial categories [35].

### 2.3. Genotyping of Virulence Factors

DNA was extracted from the overnight culture of all the isolates using bacterial genomic DNA Kit (Sigma Aldrich, St. Louis, MO, USA). All the thirty-four isolates were genotyped for virulence genes, such as genes involved in T3SS, *exoS*, *exoT*, *exoU*, *exoY*, *pscL*, *pscU,* elastase *lasB*, proteases like *aprA*, and *prpL* and a gene involved in biofilm formation, *ladS*. The PCR was performed using KAPA Taq ReadyMix with dye (KAPA Biosystems, Sigma Aldrich, St. Louis, MO, USA) using the following conditions for all except *pscU* and *pscL* denaturation at 95 °C for 30 s, annealing at 60 °C for 30 s and extension at 72 °C for 30 s for 30 cycles. *pscU* and *pscL* were amplified as previously described [30]. Table 1 lists the sequences of the primer used for amplification of the genes. PAO1 that produces *exoS, exoY and exoT* and PA14 strain that expresses all the exotoxins were used as controls.

### 2.4. Biofilm Assay

Biofilm formation was estimated using the crystal violet assay [36]. Fresh overnight cultures of the isolates were diluted to 1:100 in a 96-well plate and incubated in a shaker incubator for 24 h. The absorbance of the bacterial cultures was recorded at 600 nm prior to the start of the assay. For the assay, the wells were washed with distilled water; the biofilms were fixed using 95% methanol for 15 min followed by staining with 0.5% of crystal violet for 10 min. After washes, the dye was dissolved in 30% acetic acid and the absorbance was measured at 590 nm. The OD_590_ values were then normalized with initial OD_600_ values to account for differences in bacterial growth. Biofilm was classified as weak, moderate or strong as previously described [37]. Cut-off OD (ODc) was defined as 3 standard deviations more than the average OD of the blank. Isolates with OD < ODc were considered as non-biofilm producers, with ODc < OD < 2ODc as weak biofilm producers, 2ODc < OD < 4ODc as moderate biofilm producers and OD > 4ODc as strong biofilm producers.

### 2.5. Swarming Assay

The swarming motility of *P. aeruginosa* isolates were examined according to the protocol described earlier [27]. Briefly, a single colony was inoculated on swarming media (Bacteriological agar-0.5%, Nutrient broth-8g/L, Glucose-5g/L) and incubated overnight at 37 °C. The plates were imaged and analysed using Image J software [38]. The swarming motility was assessed as percentage change with respect to PA01 as described elsewhere [28]. An isolate showing a change of more than 10% compared to PA01 was considered as good swarmer while the rest were categorized as poor swarmer.

### 2.6. Pyoverdine Estimation

Pyoverdine production was estimated as previously described [39]. Briefly, the absorbance of overnight cultures of each isolate was recorded at 600 nm before the cultures were centrifuged at 10,000× *g* for 2 min and the absorbance of the supernatant was recorded at 405 nm. OD_405_ was normalized using OD_600_ to account for differences in bacterial growth. The normalized absorbance reading was used to estimate pyoverdine concentration as follows: Molar concentration = Absorbance/Extinction coefficient (1.9 × 10^4^ M^−1^ cm^−1^) [39]. Isolates showing a change of more than 10% compared to PA01 were considered as good pyoverdine producers, while the rest were categorized as poor producers of pyoverdine.

### 2.7. Culture of HCEC

Immortalized human corneal epithelial cells (HCEC) 10.014 pRSV-T [31,40] were maintained in DMEM-F12 media supplemented with 10% fetal bovine serum, 4 µg/mL recombinant human insulin (Invitrogen, Waltham, MA, USA) and 20 ng/mL recombinant human epidermal growth factor (Invitrogen, MA, Waltham, USA) at 37 °C and 5% CO_2_ and cultured as mentioned before [31].

### 2.8. Cytotoxicity Assay

Cell-based cytotoxicity was examined in four clinical isolates namely LVP3 (*exoS*+/*exoU*−), LVP27 (*exoS*−/*exoU*+), LVP30 (*exoS*−/*exoU*−) and LVP40 (*exoS*+/*exoU*+) along with the MDR and XDR isolates. HCEC (2.5 × 10^4^ cells/well) were seeded in a 96-well plate for lactate dehydrogenase (LDH) cytotoxicity assay. The cells were infected with each of the clinical isolates, PAO1 and PA14 at multiplicity of infection 10 (MOI, bacteria: cells 10:1) for 6 h. The culture supernatant was used for LDH estimation [41] by colorimetric assay using the CytoTox96 kit (Promega, Madison, WI USA).

### 2.9. Statistical Analysis

Bar graphs and error bars represent the mean and the standard error of mean (SEM) respectively. Statistical analysis was performed using either Kruskal-Wallis or unpaired *t* test (Prism; GraphPad Software, San Diego, CA, USA). The correlations were calculated using Spearman’s correlation test. *p* values less than 0.05 were considered significant.

## 3. Results

### 3.1. Clinical Features

Ocular clinical isolates of *P. aeruginosa* collected from thirty four patients were evaluated in the current study. The age of the patients ranged from 21 to 84 years (mean, 45.39 ± 3.19 years). There were 24 male patients and 10 female patients, and 30% of all patients were involved in agriculture, whereas 20% worked as manual laborers, and the remaining 50% were either office workers, students or homemakers. None of the patients were currently wearing or had a history of wearing contact lenses. The size of the hypopyon ranged from <1 mm to 5.8 mm and the size of the epithelial defect ranged from 2 × 2 mm to 10 × 9.5 mm. However, the size of the epithelial defect was not associated with the treatment outcome. A total of six patients underwent corneal grafting and three underwent evisceration. One of the patients who underwent corneal grafting was infected with an MDR strain and another one who underwent evisceration was infected with an XDR strain.

### 3.2. Antibiotic Susceptibility of the Clinical Isolates

The antibiotic susceptibility of the isolates was tested by utilizing the minimum inhibitory concentration (MIC) method. A total of twenty antibiotics were tested on these isolates and details are shown in Table 2. Three out of the thirty-four isolates were MDR and four isolates were found to be XDR in nature. Resistance was noted to chloramphenicol (*n* = 30), ciprofloxacin (*n* = 6), to moxifloxacin (*n* = 28), piperacillin/tazobactum (*n* = 17), ticarcillin/clavulanic acid (*n* = 16), levofloxacin and ceftazidime (*n* = 13), gatifloxacin (*n* = 7), ofloxacin (*n* = 7), gentamicin (*n* = 6), amikacin (*n* = 5), tobramycin (*n* = 6), cefepime (*n* = 5), imipenem (*n* = 5), doripenem (*n* = 4), meropenem (*n* = 4), All isolates were resistant to tigecycline. Intermediate resistance was also noted to chloramphenicol (*n* = 2), ciprofloxacin (*n* = 4), moxifloxacin (*n* = 2), cefepime (*n* = 6), doripenem (*n* = 3), meropenem (*n* = 1), piperacillin/tazobactum (*n* = 5), ticarcillin/clavulanic acid (*n* = 15) and cefoperazone (*n* = 15). All the isolates were however susceptible to colistin. A heat map was constructed depicting the relative resistance of each isolate (Figure 1).

### 3.3. Differential Expression of T3SS Genes among the Clinical Isolates of P. aeruginosa

*P. aeruginosa* has a repertoire of toxins secreted by different secretory pathways. T3SS is one of the major virulence factors that have been shown to subvert host immune responses, including reactive oxygen species generation, in human corneal epithelial cells [42,43]. In this study, we screened thirty four ocular clinical isolates causing corneal infections and determined the presence of genes associated with virulence such as the main T3SS effector genes, *exoS*, *exoT*, *exoU* and *exoY*, as well as *pscU* and *pscT*, responsible for T3SS machinery [44]. The presence of *exoS* is associated with increased invasiveness and presence of *exoU* is associated with increased cytotoxicity [4]. As shown in Figure 2A, 73% of the clinical isolates encoding *exoS* were invasive, whereas about 32% of the isolates were cytotoxic with the presence of *exoU* gene. Approximately 85% of the isolates showed presence of *exoY*. Interestingly, all the three genes were present in only 12% of the isolates and were completely absent in 9% of the isolates. *exoT*, associated with T3SS apparatus, was present in all the isolates while *pscU* and *pscT* were present in all isolates except one each. The presence of *exoS* and *exoU* was mutually exclusive, as reported before, except in four isolates which showed presence of both the genes [45]. Moreover, a majority of the MDR and XDR strains harbored *exoU* gene further suggesting an increased cytotoxicity of these strains. However, 78% of patients that underwent penetrating keratoplasty were infected with isolates harboring *exoS*. Along with the exotoxins, *P. aeruginosa* also produces several extracellular proteases of which alkaline proteases (*aprA*), elastase B (*lasB*) and protease IV (*prpL*) are often implicated in infections and helps the pathogen in immune evasion [46]. All the isolates investigated in this study were found to be positive for the genes *lasB, aprA* and *prpL*. All of these proteases have also been found to play an important role in corneal damage during *Pseudomonas* keratitis [15]. Our data are in concordance with two earlier results carried out on clinical and environmental isolates indicating these genes to be universally present [45,47]. We also examined the presence of *ladS*, a calcium binding kinase that promotes biofilm formation on activation [20], and found it to be present in all the isolates. A heat map was constructed depicting the presence of all the genes of each isolate (Figure 2B).

### 3.4. Biofilm Assay

The resistance of *P. aeruginosa* against antibiotics also results from its ability to form biofilm; therefore, we determined the ability of biofilm formation in these isolates. According to the classification that we followed, all the isolates were capable of forming biofilms at different levels. Sixteen percent of the isolates formed weak biofilms, 50% formed moderate biofilms and the remaining 35% were strong biofilm formers (Figure 2C). Biofilms formed by *P. aeruginosa* are known to be resistant to antibiotics, and all the multi-drug resistant clinical isolates identified in this study are moderate to strong biofilm producers. We did not observe an effect of T3SS genotype on biofilm formation; however, around 65% of the strong-moderate biofilm formers showed the presence of *exoS*.

### 3.5. Swarming Motility is Linked to Biofilm Formation

*P. aeruginosa* swarming is a complex adaptation process influenced by major changes in gene expression, involves multicellular coordination and exhibits a strong interrelation with biofilm formation. Several regulatory pathways responsible for swarming also affect the formation of biofilm [48]. In our study we found that 68% of the isolates that formed moderate to weak biofilm were good swarmers. The swarming activity and biofilm formation of the isolates were negatively correlated (Spearman’s correlation coefficient: r = −0.3742, *p* = 0.0268) (Figure 2D). This also correlates well with previous studies which have found an inverse relationship between biofilm formation and swarming motility [28,48]. Another virulence related gene, *lasB*, is known to play an important role in swarming [12]. All our isolates were, however, positive for the presence of *lasB* gene irrespective of their swarming ability. Enhanced antibiotic resistance has been reported in swarmer cells of *P. aeruginosa* [12]; however, we did not see any such association. Interestingly, we found that about 79% of the good swarmers did not harbor *exoU* suggesting that environmental cues might facilitate selection of either swarming motility or cell cytotoxicity.

### 3.6. Pyoverdine Secretion among Isolates

A recent study by Suzuki et al. demonstrated the importance of pyoverdine production in *Pseudomonas* corneal infection using a mouse model of keratitis [49]. We estimated pyoverdine production in overnight cultures of all the clinical isolates and demonstrated no significant difference in the concentration of pyoverdine synthesized by these isolates. Sixty nine percent of the isolates were found to produce more pyoverdine than PAO1, and this was not associated with their T3SS genotype. Pyoverdine concentration was significantly different among various groups of biofilm forming isolates (Figure 2E). We found positive correlation between biofilm formation and pyoverdine secretion of the isolates (Spearman’s correlation coefficient: r = 0.4173, *p* = 0.0141). No correlation was found in a recent study between biofilm formation and pyoverdine production among various clinical and environmental isolates [50]. *pvdQ*, gene responsible in pyoverdine biosynthesis, has also been shown to play an important role in swarming. Overhage et al. reported increased expression of *pvdQ* gene in PA14 under swarming condition [12]. Although we did not find any direct correlation between swarming and pyoverdine secretion (Figure 2F), many of the isolates that secreted higher concentration of pyoverdine were good swarmers.

### 3.7. T3SS Positive Isolates Caused Increased Cell Death in HCEC

We performed cell-based assays to determine the cytotoxicity towards human corneal epithelial cells of a few selected isolates. For this purpose, we chose four isolates depending on their genotype, LVP3, LVP27, LVP30 and LVP40, along with the MDR (LVP29 and LVP39) and XDR (LVP25, LVP33, LVP35, LVP41) isolates, and laboratory strains PAO1 and PA14 were used as controls. LVP3, LVP27 and LVP40 showed increased cytotoxicity and were comparable to that of PAO1 and PA14 while LVP30 was least cytotoxic to the cells (Figure 3A). We also checked the cytotoxicity of corneal epithelial cells with the drug resistant isolates and found increased cytotoxicity comparable to PAO1 (Figure 3B). The isolates LVP3, 27 and 40 show presence of either *exoS* or *exoU* and exhibited increased cytotoxicity, whereas an XDR isolate, LVP25 (*exoS−/exoU+*) was comparably less cytotoxic. Out of the ten isolates tested, three isolates, LVP30, 33, and 39 lack both exoS and exoU, and interestingly while LVP 30 showed reduced cytotoxicity, LVP 33 and 39 were cytotoxic to cells. These results suggest that perhaps T3SS is not the only determinant of the damage caused by bacteria to cells.

## 4. Discussion

*P. aeruginosa,* a versatile, opportunistic pathogen causes corneal infections that are often difficult to treat due to emergence of antibiotic resistance and multi-drug resistant isolates are often encountered in the clinic. In this study we examined the different virulent characteristics of ocular clinical isolates causing infections to understand their role during pathogenesis of disease.

T3SS is a well-established mode of virulence for *P. aeruginosa* and plays a prominent role in causing infections. The four effector proteins of T3SS that are involved in virulence include ExoS, ExoU, ExoT and ExoY. *exoS* and *exoT* encode for bifunctional enzymes which comprise of a GTPase activating domain and an ADP ribosyltransferase domain [51]. *exoU* encodes for a cytotoxin phospholipaseA2 and *exoY* encodes for adenylate cyclase [10]. *exoT* is known to be ubiquitously present in all the *P. aeruginosa* strains and is consistent with our study in which we found that all the isolates showed presence of *exoT* [52]. We found a higher prevalence of *exoS+* isolates than *exoU+* isolates in our cohort of strains. An earlier report on *P. aeruginosa* clinical isolates from endophthalmitis cases also showed the predominance of *exoS* positive isolates [26]. It has been previously reported that *exoU+* strains were more common in contact lens wearers [52,53], whereas our cohort of cases were non-contact lens wearers. Thus, in contrast to earlier reports [29], our data show the presence of a greater proportion of *exoS* harboring *P. aeruginosa* isolates and remains consistent with reports where they observed higher prevalence of *exoS+* isolates especially in non-contact lens wearers [30,37]. The gene expression pattern is also consistent with those observed among *P. aeruginosa* environmental isolates [45]. Sun et al. previously demonstrated that presence of *exoY* is not essential for development of keratitis [54]. A previous study comparing the virulence patterns of contact lens and non-contact lens wearers suggested that strains expressing *exoU* were stronger biofilm producers [37]. Consistently with this observation, we found that more than 77% of the isolates producing low to moderate biofilms were *exoU* negative and that more than half of the *exoU*+ isolates were strong biofilm producers. Swarming ability has also been negatively correlated with biofilm formation [28,48]. Our results also show that out of all the positive swarmers, 79% were low to moderate biofilm formers. Perhaps the bacteria require one or the other characteristic for colonization in the host cornea.

The presence of *exoU* has also been reported to lead to increased resistance against antibiotics especially fluoroquinolones [55,56,57]. Interestingly, out the thirty-four isolates screened in this study, seven were resistant to fluoroquinolones and 85% were *exoU*+. In a recent study by Horna et al., out of 189 *P. aeruginosa* isolates obtained from patients in the intensive care unit, majority of the multi-drug and extensively-drug resistant strains were *exoU* positive [58]. In agreement with this, we found that out of the seven strains which were multi-drug resistant, five were *exoU* positive. Moreover, these MDR isolates formed moderate to strong biofilms which is consistent with association of biofilm formation with antibiotic resistance reported previously [19]. Swarming has also been associated with increased antibiotic resistance [12]. However, in the present study, we did not observe any such relationship; more than half of the MDR isolates were poor swarmers. Possibly, the bacteria adapt to different behavior to confer antibiotic resistance and the MDR isolates in this study were better at biofilm formation than swarming. Swarming is a complex phenomenon of motility of bacteria over soft surfaces and swarming cells of *P. aeruginosa* exhibit different phenotype from planktonic cells including gene expression [59] and antibiotic resistance [60]. There are several reports regarding the inverse relationship between biofilm formation and swarming motilities [61,62] and this regulation is mediated by cyclic diguanylate that induce biofilm formation and suppresses swarming. Murray et al. observed a similar inverse relationship between swarming and biofilm formation for a large cohort of 237 non-ocular clinical isolates [28]. Inverse regulation of biofilm formation and swarming has also been reported earlier for PA14 strain [62].

Pyoverdine regulates several virulent factors and plays a critical role in the pathogenesis of host infection by *P. aeruginosa* [63]. It removes ferric iron from the host causing mitochondrial damage and compromise ATP production [64]. Suzuki et al. have shown that a pyoverdine mutant strain is incapable of invading corneal epithelial cells and fails to cause infection in a murine model of keratitis compared to PAO1 [49]. Pyoverdine aids in colonization to the host and promotes biofilm formation. Consistent with this, in the present study we observed a positive association with biofilm formation, however we did not observe an effect of the T3SS genotype. A recent study by Kang et al. did not show any correlation between biofilm formation and pyoverdine production among various clinical and environmental isolates, although they found a positive correlation among low biofilm forming subsets [50]. *pvdQ*, gene responsible in pyoverdine biosynthesis, has also been shown to play an important role in swarming. Overhage et al. reported increased expression of *pvdQ* gene in PA14 under swarming condition [12]. Although we did not find any direct correlation between swarming and pyoverdine secretion, many of the isolates that secreted higher concentration of pyoverdine were also good swarmers.

Isolates from different origins are associated with different virulence factors [37]. To further examine the cytotoxicity of the isolates we chose four isolates based on their T3SS genotype. The presence of *exoS* is associated with increased invasiveness and presence of *exoU* is associated with increased cytotoxicity [4]. *exoU+* isolates have previously been reported to mediate pathogenicity in an experimental model of keratitis and induce cell lysis in macrophages and epithelial cells. Seventy eight percent of patients undergoing corneal transplantation in this cohort were infected with *exoS+* isolates. However, we found from the cytotoxicity assays that the T3SS profile was not the only determinant of cell cytotoxicity and other virulence factors might also contribute to cell damage. Consistent with this, Toska et al. reported that few T3SS negative *Pseudomonas* were still capable of causing infection in a murine model of keratitis [29]. We examined the presence or absence of the effector genes and a further investigation of the expression of these effector proteins in culture or in an infection model will confirm the T3SS expression profile. A recent report by Hwang et al. showed reduced virulence of multi-drug resistant isolate of *P. aeruginosa* in a mouse model of lung infection [65], however we found increased cytotoxicity of MDR and XDR isolates in corneal epithelial cells in vitro. Moreover, even the strains which do not show presence of *exoS* and *exoU* may have developed a novel virulence pathway which facilitates the colonization in the host and may be a subject for further investigation.

In conclusion, we extended the understanding of the virulence factors and other characteristics of ocular clinical isolates obtained from our cohort of patients. Overall, we found that the isolates utilized different virulence mechanisms for colonization in the host independent of gene expression pattern. The detailed understanding will perhaps assist in developing alternative therapeutic interventions to target virulence of *P. aeruginosa* without affecting its growth and will be helpful in selection of a treatment strategy.

## Figures and Tables

**Figure 1 microorganisms-08-00260-f001:**
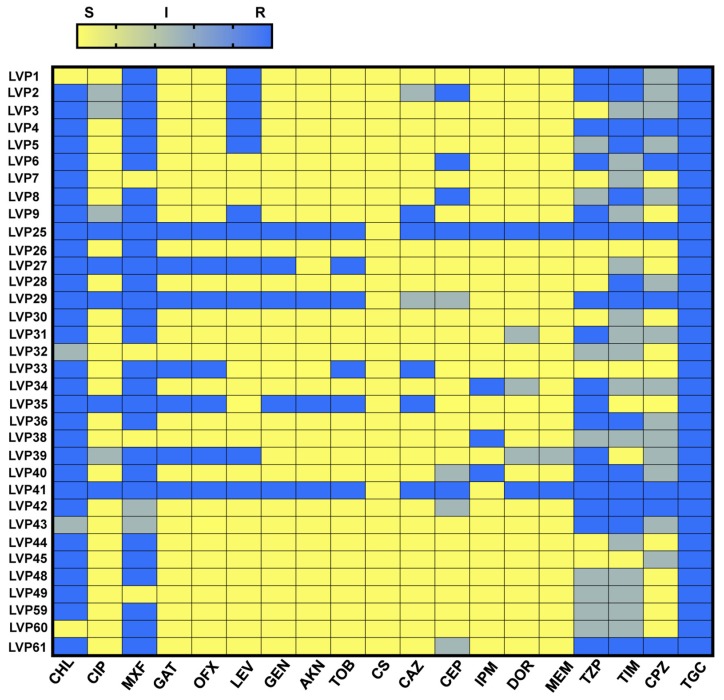
Heat map representing antibiotic resistance of ocular clinical isolates of *P. aeruginosa*. The antibiotic susceptibility for thirty-four clinical isolates were tested by minimum inhibitory concentration method against mentioned antibiotics and a heat map was constructed to compare the antibiotic resistance among the isolates. S denotes susceptible, I denotes intermediate, and R denoted resistance to antibiotics.

**Figure 2 microorganisms-08-00260-f002:**
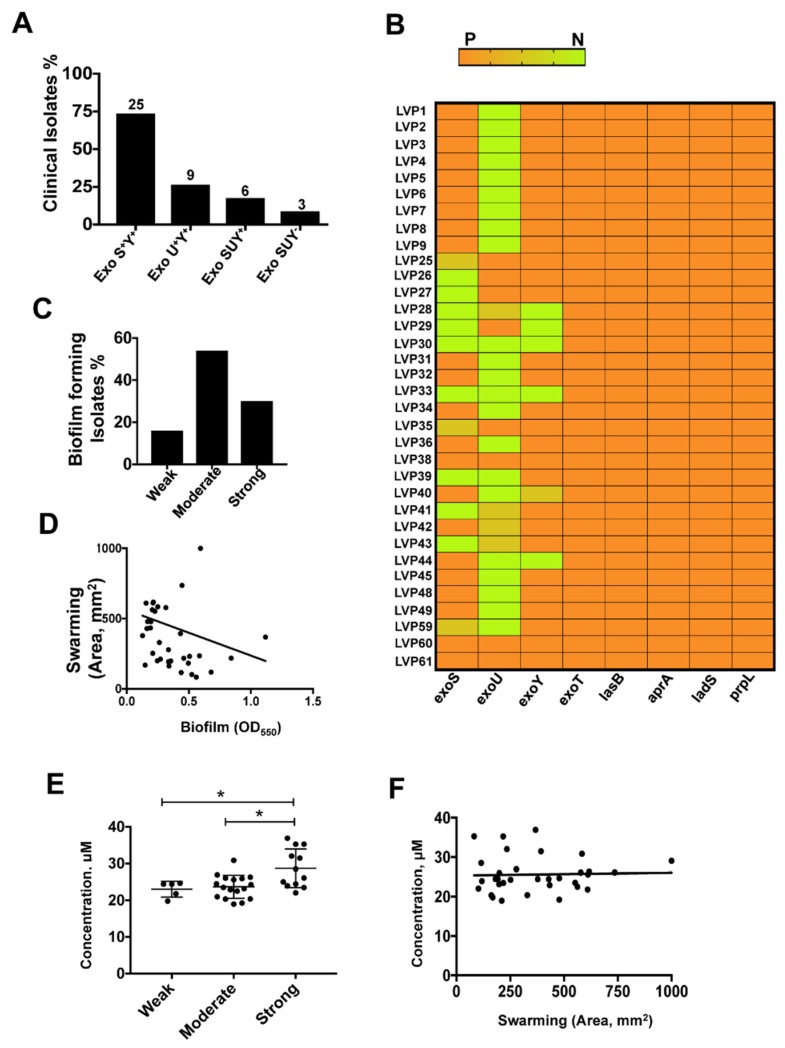
Characteristics of ocular clinical isolates of *P. aeruginosa*. The presence of T3SS effectors in ocular clinical isolates collected from patients (**A**). Heat map representing presence of virulence genes of ocular clinical isolates of *P. aeruginosa*. P denotes the presence of the gene and N denotes absence of the specific genes (**B**). Distribution of ocular clinical isolates forming biofilms; isolates were divided into weak, moderate and strong according to their biofilm forming abilities (**C**). Correlation between swarming and biofilm forming abilities (**D**), pyoverdine concentration and biofilm formation (**E**), and pyoverdine concentration and swarming (**F**) of the ocular clinical isolates. The experiments were repeated two times with similar results. * indicates *p* < 0.05.

**Figure 3 microorganisms-08-00260-f003:**
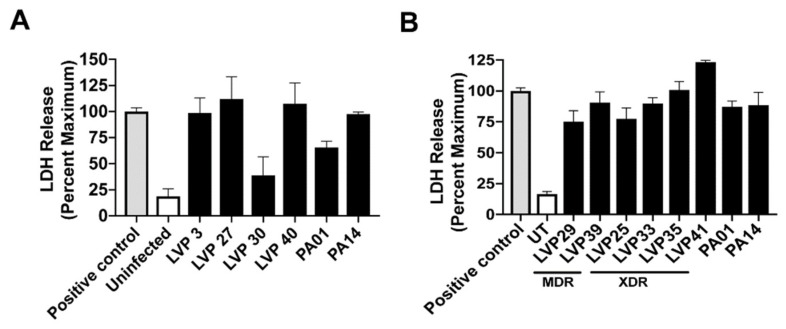
Cytotoxicity of human corneal epithelial cells by clinical isolates of *P. aeruginosa***.** Cells were infected with the clinical isolates harboring different exotoxins, LVP3 (*exoS*+/*exoU*−), LVP27 (*exoS*−/*exoU*+), LVP30 (*exoS*−/*exoU*−) and LVP40 (*exoS*+/*exoU*+) (**A**) or drug-resistant isolates (**B**) for 6h and cytotoxicity was measured by release of LDH into the culture media compared to lysed cells (positive control). The error bars represent three technical replicates and the experiments have been repeated three times. UT represents untreated cells.

**Table 1 microorganisms-08-00260-t001:** Sequences of primers used for gene amplification.

Virulence Genes (Product Length)	Primers (5′-3′)
*exoS* (235 bp)	FWD: AGAGCGAGGTCAGCAGAGTA REV: GCGGACATACCTTGGTCGAT
*exoT* (219 bp)	FWD: GCATGCGGTAATGGACAAGG REV: GACCGATTCAGGTGCTGGTA
*exoU* (134 bp)	FWD: CGGTACGTGCTGTATCCCTC REV: CGTGTAGCGCGATCTGTAGT
*exoY* (289 bp)	FWD: GCTTCTCGGTGAAGGGGAAA REV: CGAACTCATAGCGTTTGCCG
*lasB* (202 bp)	FWD: ATCGACGTGTCCAAACTCCC REV: CCTTGACTTCGGTGATGGCT
*aprA* (176 bp)	FWD: CTACAGCGCCAACGTCAATC REV: AGCTCATCACCGAATAGGCG
*ladS* (181 bp)	FWD: CCCTGATGGTCCTCGGCTAC REV: GTTCCTGGTTCAGCGCTTCC
*pscL*	FWD: AAAAAAGAATTCGGAGGGCGATGAATGCTTCCATTTGTT REV: AAAAAAAAGCTTTCAACCGGCGTCCCCTTCCTCCT
*pscU*	FWD: AAAAAATCTAGAGGAGGAGACGCCATGAGCGCCGAGAAGA REV: AAAAAAAAGCTTGATAGCGATCAGGGCGTATCCGTCTGCT
*prpL*	FWD: ATCGTATTTCGCCGACTCCC REV: TGAAGACCATCTTCGCCACC

**Table 2 microorganisms-08-00260-t002:** Minimum Inhibitory Concentration based Antibiotic Susceptibility Pattern of Ocular Clinical Isolates of *P. aeruginosa* (*n* = 34).

Antibiotic	MIC (μg/mL)	% Isolates		
		Susceptible (S)	Intermediate (I)	Resistant (R)

Chloramphenicol (CHL)	0.016–256	6	6	88
Ciprofloxacin (CIP)	0.25–4	70	12	18
Moxifloxacin (MXF)	0.002–32	12	6	82
Gatifloxacin (GAT)	0.002–32	79	0	21
Ofloxacin (OFX)	0.002–32	79	0	21
Levofloxacin (LEV)	0.12–8	62	0	38
Gentamycin (GEM)	1–16	82	0	18
Amikacin (AKN)	2–64	85	0	15
Tobramycin (TOB)	0.016–256	82	0	18
Colistin (CS)	0.5–16	100	0	0
Ceftazidime (CAZ)	1–64	62	0	38
Cefepime (CEP)	1–64	67	18	15
Imipenem (IPM)	0.25–16	85	0	15
Doripenem (DOR)	0.12–8	79	9	12
Meropenem (MEM)	0.25–16	85	3	12
Piperacillin/Tazobactam (TZP)	4/4/–128/4	35	15	50
Ticarcillin/Clavulanic Acid (TIM)	8/2–128/2	9	44	47
Cefoperazone/Sublactam (CPZ)	8–64	32	44	24
Tigercycline (TGC)	0.5–8	0	0	100

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
