# Peer review of "Characterization of Ocular Clinical Isolates of Pseudomonas aeruginosa from Non-Contact Lens Related Keratitis Patients from South India"

_microorganisms, 2020, doi:10.3390/microorganisms8020260_

Round 1

Reviewer 1 Report

If the data is available for the presence / absence of protease IV in your strains, i would consider adding it to the paper

Author Response

If the data is available for the presence / absence of protease IV in your strains, i would consider adding it to the paper.

Response: We thank the reviewer for this comment. We have now added the data for protease IV in results section (highlighted) and in figure 2 as 2B.

Reviewer 2 Report

Comment #1

what is the rationale to mention “30% of all patients were involved in agriculture, whereas 20% worked as manual laborers, office workers or as homemakers”? What about other 50% of the patients. Does this have any correlation with results? It is important to mention that all the patients are non-contact lens wearers but if above mentioned data has no effect on the outcome of the results, then it is irrelevant information in my opinion.

Comment #2

In section 2.4, the way it is written seems like they measured OD590 and OD600 at the same time. It would be beneficially if they clarify that like they did in section 2.6

Comment #3

The way the results section 3.3 currently written is very confusing, the authors should rather provide that data in a table format or something like Figure 1. with genes in place of antibotics and + or – in place of heat map would be much beneficial in my opinion.

Comment # 4

The authors have provided the details of selected genes in section 3.3 but it improves the flow and clarity of the manuscript if they provide that information prior in introduction or materials section.

Comment #5

What passage of human corneal epithelial cells did the authors used? What is the source of these cells? Were all the replicates for cytotoxicity studies done on same passage of cells?

Comment #6

Deducting T3SS positive isolates cause increased cell death based one patient isolate that is T3SS negative is not apt in my opinion. Of the 34 samples that the authors have is this the only sample with exoS-/exoU-. If not, why didn’t the authors used more samples.

Author Response

Comment #1

What is the rationale to mention “30% of all patients were involved in agriculture, whereas 20% worked as manual laborers, office workers or as homemakers”? What about other 50% of the patients. Does this have any correlation with results? It is important to mention that all the patients are non-contact lens wearers but if above mentioned data has no effect on the outcome of the results, then it is irrelevant information in my opinion.

Response: We thank the reviewer for this comment. There was an omission in the statement and we apologize for that. It should have read, “and 30% of all patients were involved in agriculture, whereas 20% worked as manual laborers, and the remaining 50% were either office workers, students or homemakers” and this has now been added and highlighted.We added the data as a part of patient’s demographics and to emphasis that corneal infection can occur across all socio economic spectrum and people with different professions.

Comment #2

In section 2.4, the way it is written seems like they measured OD590 and OD600 at the same time. It would be beneficially if they clarify that like they did in section 2.6

Response: We have now corrected it in the text. Thank you.

Comment #3

The way the results section 3.3 currently written is very confusing, the authors should rather provide that data in a table format or something like Figure 1. with genes in place of antibotics and + or – in place of heat map would be much beneficial in my opinion.

Response: We thank the reviewer for this comment. A heat map depicting the data has now been included as figure 2B.

Comment # 4

The authors have provided the details of selected genes in section 3.3 but it improves the flow and clarity of the manuscript if they provide that information prior in introduction or materials section.

Response: We thank the reviewer for this comment. We have now added the details in the introduction (highlighted).

Comment #5

What passage of human corneal epithelial cells did the authors used? What is the source of these cells? Were all the replicates for cytotoxicity studies done on same passage of cells?

Response: The cells used for the experiment are from passage number 45-49. The source of the cells is now included in the text, reference [1]. The replicates were not from the same passage of cells. The results from these experiments were reproducible which indicates that the clinical isolates lead to increased cytotoxicity regardless of the different passages of the cells used in the study.

Comment #6

Deducting T3SS positive isolates cause increased cell death based one patient isolate that is T3SS negative is not apt in my opinion. Of the 34 samples that the authors have is this the only sample with exoS-/exoU-. If not, why didn’t the authors used more samples.

Response: We apologize for the extrapolation of results. We re-analysis of our results, there are three isolates (LVP30, LVP33 and LVP39) which lack both exoSand exoU. While LVP30 is less cytotoxic, the other two isolates showed higher cytotoxicity towards epithelial cells. Therefore, the T3SS machinery may not be the only factor inducing cytotoxicity cells. We have now clarified this in the updated manuscript as well. 

Round 2

Reviewer 2 Report

The Authors have answered all my concerns. However, I have following comment for them to improve the manuscript.

In both the heat maps that authors used 1-34 a samples id however, in section 3.7 they started used LVP3, 27, 30 39 etc. To be consistent they should use same sample id’s through out the manuscript in my opinion.

Author Response

Reviewer 2

Comment 1: The Authors have answered all my concerns. However, I have following comment for them to improve the manuscript.

In both the heat maps that authors used 1-34 a samples id however, in section 3.7 they started used LVP3, 27, 30 39 etc. To be consistent they should use same sample id’s through out the manuscript in my opinion.

Response: We thank the reviewer and apologise for the inconsistency in numbering. We have now corrected it and same sample id is reflected all over.

We thank the editor and reviewers for helping us to improve the manuscript.

With regards,

Sanhita

10-02-20